

# Dynamic multiple-graph spatial-temporal synchronous aggregation framework for traffic prediction in intelligent transportation systems

Xian Yu[1,2], Yinxin Bao[1] and Quan Shi[1,3]

[1] School of Information Science and Technology, Nantong University, Nantong, Jiangsu, China
[2] Xinglin College, Nantong University, Nantong, Jiangsu, China
[3] School of Transportation and Civil Engineering, Nantong University, Nantong, Jiangsu, China

## ABSTRACT

Accurate traffic prediction contributes significantly to the success of intelligent transportation systems (ITS), which enables ITS to rationally deploy road resources and enhance the utilization efficiency of road networks. Improvements in prediction performance are evident by utilizing synchronized rather than stepwise components to model spatial-temporal correlations. Some existing studies have designed graph structures containing spatial and temporal attributes to achieve spatial-temporal synchronous learning. However, two challenges remain due to the intricate dynamics: (a) Accounting for the impact of external factors in spatial-temporal synchronous modeling. (b) Multiple perspectives in constructing spatial-temporal synchronous graphs. To address the mentioned limitations, a novel model named dynamic multiple-graph spatial-temporal synchronous aggregation framework (DMSTSAF) for traffic prediction is proposed. Specifically, DMSTSAF utilizes a feature augmentation module (FAM) to adaptively incorporate traffic data with external factors and generate fused features as inputs to subsequent modules. Moreover, DMSTSAF introduces diverse spatial and temporal graphs according to different spatial-temporal relationships. Based on this, two types of spatial-temporal synchronous graphs and the corresponding synchronous aggregation modules are designed to simultaneously extract hidden features from various aspects. Extensive experiments constructed on four real-world datasets indicate that our model improves by 3.68–8.54% compared to the state-of-the-art baseline.

## INTRODUCTION

Intelligent transportation systems (ITS) provide efficient guidance for real-time traffic management and assist people in scheduling their travel plans in advance (*Shaygan et al., 2022*). An essential function of ITS is traffic prediction, based on which it can optimize the allocation of road network resources and reduce traffic problems such as congestion and accidents (*Kong et al., 2024*). Therefore, the operation of ITS is heavily dependent on precise traffic prediction, the core of which is modeling spatial-temporal dynamics of

Corresponding author
Quan Shi, sq@ntu.edu.cn

traffic features. Recent years have witnessed a widespread application of graph convolutional network (GCN) for extracting spatial correlations, where the distribution of traffic sensors is modeled as a series of nodes and edges in a graph (*Bao et al., 2023*; *Kong et al., 2022*; *Chen et al., 2022*; *Huang et al., 2022*). In addition, recurrent neural network (RNN) and its variants, also known as long short-term memory (LSTM) and gated recurrent unit (GRU) have been extensively applied to model temporal dependency due to their outstanding performance in processing time series (*Zhao et al., 2023*; *Ma et al., 2023*; *Afrin & Yodo, 2022*; *Ma, Dai & Zhou, 2022*). Some studies employ convolutional neural network (CNN) instead of RNN to learn temporal dynamics (*Wen et al., 2023*; *Ni & Zhang, 2022*). To synchronize the extraction of spatial-temporal features, some work has designed graph structures that contain both spatial and temporal attributes (*Song et al., 2020*; *Li & Zhu, 2021*; *Jin et al., 2022*; *Wei et al., 2023*). In spite of the pioneering advances in these studies, there is still a lack of sufficiently practical approaches in spatial and temporal synchronous learning owing to the complexity of traffic dynamics.

Firstly, traffic features depend not only on their historical data but are also influenced by external factors. As illustrated in Fig. 1A, the traffic flow varies significantly with meteorological factors. For example, in the case of sunny weather and comfortable temperatures, there is a significant increase in traffic flow in the tourist area. In contrast, there is a remarkable decrease in the same area in the case of heavy rain and cold environments. Secondly, there are multiple spatial-temporal relationships between traffic nodes. Spatially, different nodes can be measured by neighborhood or distance. Temporally, the traffic flow between different nodes shows the same pattern or exhibits the same trend. As shown in Fig. 1B, nodes A and B's traffic flow rise between 5:00 and 10:00 and fall between 17:00 and 22:00, and their rising and falling rates are almost the same, showing a strong linear correlation, thus confirming that they have the same pattern. The traffic flow of node C, although it has a similar variation interval, alters more gently and shows a weaker linear correlation with the flow of node A. Therefore, they exhibit the same trend despite their different patterns. For these reasons, two challenges remain in learning spatial-temporal dependencies of traffic features.

(a) Accounting for the impact of external factors in spatial-temporal synchronous modeling. Traffic features change in response to external factors, so learning the effects of external factors in spatial-temporal synchronous modeling is necessary. *Zhu et al. (2021)* introduced dynamic and static external factors and then encoded them into a graph convolutional network to obtain predictions that take external factors into account. The study by *Qi et al. (2022)* proposed an attribute feature unit to fuse weather conditions, temperature, visibility, as well as traffic flow, and fed the fused features into temporal graph convolutional network (T-GCN) for modeling spatial-temporal dependencies. *Sun et al. (2022)* utilized interactive and internal attention mechanisms to embed traffic data and external factors into high-dimensional sequences, which increased the accuracy of predictions. These studies took into consideration the influence of external factors, but they failed to accomplish spatial-temporal simultaneous modeling. In contrast, spatial-temporal synchronous graph convolutional networks (STSGCN) (*Song et al., 2020*), spatial-temporal fusion graph neural networks (STFGNN) (*Li & Zhu, 2021*), automated

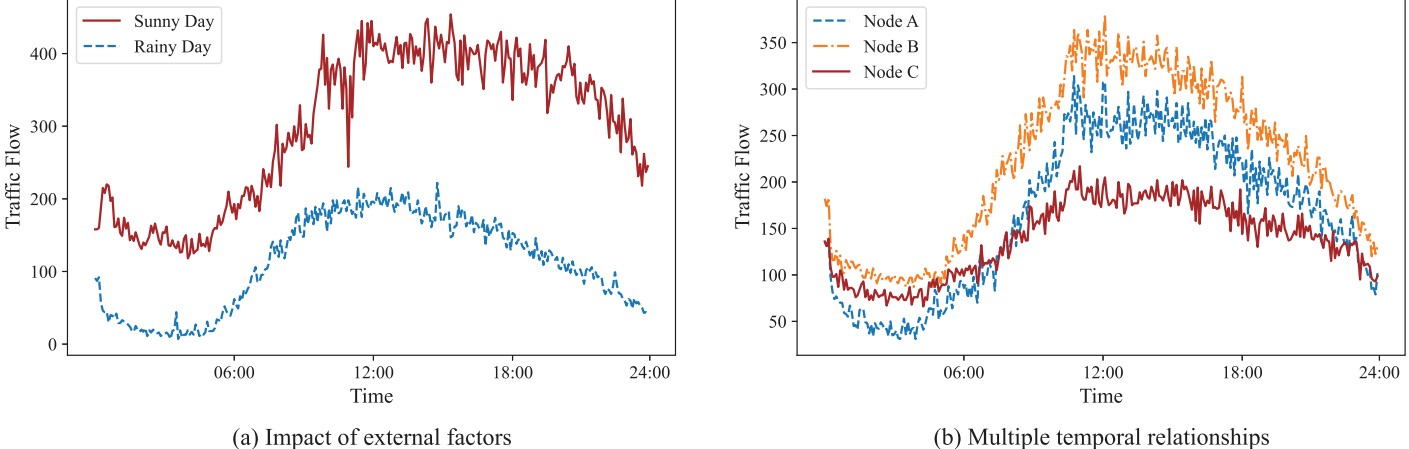

(a) Impact of external factors          (b) Multiple temporal relationships

**Figure 1** **Examples of the impact of external factors and the multiple temporal relationships.** (A) Graph showing that the traffic flow on a sunny day is distinctly different from that on a rainy day. (B) Graph illustrating that nodes A and B, with a strong linear correlation, have the same pattern, while nodes C and A exhibit the same trend. 

dilated spatio-temporal synchronous graph network (Auto-DSTSGN) (*Jin et al., 2022*) and spatial-temporal graph synchronous aggregation model (STGSA) (*Wei et al., 2023*) synchronously learned spatial and temporal dynamics. However, they only adopted traffic features as inputs to the model and ignored the impact of external factors. Overlooking external factors in spatial-temporal synchronous modeling causes obvious deficiencies in traffic prediction.

(b) Multiple perspectives in constructing spatial-temporal synchronous graphs. Both spatial and temporal relationships of traffic features are intricate and can be characterized through various perspectives. Modeling spatial and temporal dependencies from a single aspect can result in the neglect of some vital information. STSGCN (*Song et al., 2020*) built localized synchronous graphs according to whether nodes are adjacent in time or space. Based on this, STFGNN (*Li & Zhu, 2021*) introduced temporal graphs produced by a dynamic time warping algorithm to model temporal dependency. Further, Auto-DSTSGN (*Jin et al., 2022*) reduced the size of synchronous graphs while keeping the spatial and temporal graphs consistent. STGSA (*Wei et al., 2023*) proposed heuristic spatial graphs, but its synchronous graphs are limited to simple connections in the time dimension. Learning spatial or temporal correlation from only one perspective limits existing models' capabilities to extract hidden features.

To address the mentioned limitations, a novel model called dynamic multiple-graph spatial-temporal synchronous aggregation framework (DMSTSAF) is proposed in this article. Specifically, we design a feature augmentation module (FAM) that employs spatial and temporal attention mechanisms to integrate traffic features and external factors. In addition, we construct multiple spatial-temporal synchronous graphs to model various spatial-temporal dynamics. Our main contributions to this work are as follows:

- We propose a feature augmentation module to combine traffic features with external factors, in which spatial and temporal attention mechanisms are applied to integrate the two inputs adaptively and generate fused features.
- We construct diverse spatial and temporal graphs, and consequently design two kinds of dynamic spatial-temporal synchronous graphs and the corresponding synchronous aggregation modules, which model spatial and temporal correlations simultaneously in multiple perspectives.
- To test the performance of our model in diverse cases, we conduct extensive experiments on four real-world datasets. The numerical results indicate that DMSTSAF improves by 3.68–8.54% in comparison to the state-of-the-art baseline, demonstrating the consistent superiority of the proposed model.

The rest of this article is organized as follows. "Literary Review" provides the related work on graph neural network and traffic prediction. "Preliminary" introduces the mathematical definition of the task. "Methodology" presents the detailed process of our methodology. In "Experiments", extensive experiments are demonstrated, and the results are analyzed. "Discussion" states the concluding remarks.

## LITERARY REVIEW

### Graph neural network

Compared to CNN, graph neural network (GNN) can handle non-Euclidean data, which has led to its wide application in many fields such as feature extraction, node classification, *etc*. The classifications of GNN are categorized into two types: spectral domain and spatial domain. The former, also known as GCN, has undergone three important developments. *Bruna et al. (2014)* implemented graph convolution operations by replacing the convolution kernel with a learnable diagonal matrix. To overcome the problem of excessive computation, *Defferrard, Bresson & Vandergheynst (2016)* introduced Chebyshev polynomials to approximate the convolutional kernel. *Kipf & Welling (2017)* further reduced Chebyshev polynomials to the first order, which greatly simplified the computation and obtained the most common GCN expressions. The latter aims to define GNN by iteratively updating the representation of nodes from spatial neighbor aggregation. In order to avoid excessive nodes participating in the computation, graph sample and aggregate (GraphSAGE) *et al.* (*Hamilton, Ying & Leskovec, 2017*) limited the number of neighboring nodes by sampling and then achieved information aggregation through pooling operations. *Atwood & Towsley (2016)* defined the weights of neighbors using the K-hop transfer probabilities obtained after a random walk. Graph attention network (GAT) (*Velivčkovič et al., 2018*) employed the attention mechanism to define the weights for various neighbors, which allowed for a more flexible characterization of the aggregation in different scenarios.

### Traffic prediction

In recent years, neural network has been extensively utilized for traffic prediction, which offers superior modeling performance compared to traditional methods (*Guo et al., 2021*).

Most studies utilized graph convolution network to capture spatial feature (*Li et al., 2022*; *Wang et al., 2022*; *Zhu et al., 2022*; *Yao et al., 2023*; *Yang et al., 2022*). GraphSAGE has also been used to model spatial dependency for inductive learning (*Liu et al., 2023*; *Liu, Ong & Chen, 2022*). Temporal modules based on RNN and its LSTM, as well as GRU, have been introduced to learn temporal dependence (*Pan et al., 2022*; *Subramaniyan et al., 2023*; *Bao et al., 2022*; *Shu, Cai & Xiong, 2022*; *Wan et al., 2022*). To improve computational efficiency, some studies employed CNN instead of RNN to model temporal correlation (*Ji, Yu & Lei, 2023*; *Zhang et al., 2022*). *Li et al. (2018)* designed an encoder-decoder architecture that employed a diffusion process characterized by a bidirectional walk of a graph to learn spatial dependency and proposed the diffusion convolutional gated recurrent unit to model temporal dynamics. Based on this, *Wu et al. (2019)* further introduced an adaptive adjacency matrix in diffusion convolution to discover hidden spatial features and then employed dilated stacked 1D convolutions with larger receptive fields to capture temporal trends. *Yu, Yin & Zhu (2018)* defined the problem on a graph and then modeled spatial correlation by utilizing graph convolution with Chebyshev polynomials approximation, and took advantage of gated CNN to extract temporal features. On this foundation, *Guo et al. (2019)* incorporated GCN and attention mechanisms to enhance the representation of features, then deployed three parallel sets of components to learn different temporal trends. T-GCN (*Zhao et al., 2020*) integrated GCN with GRU as a way to learn the complex dynamics of traffic features. Graph multi-attention network (GMAN) (*Zheng et al., 2020*) adopted exclusively attention mechanisms rather than convolutional network to transform input features into predictions, which had a high computational complexity but improved the prediction accuracy.

To achieve synchronous modeling of spatial-temporal correlations, STSGCN (*Song et al., 2020*) constructed localized synchronous graphs, which aggregated information from neighbor nodes at the current time step and from themselves at adjacent time steps. Based on this, STFGNN (*Li & Zhu, 2021*) further fused spatial graphs and temporal graphs to learn the hidden correlations simultaneously. Auto-DSTSGN (*Jin et al., 2022*) designed an automated dilated spatial-temporal synchronous graph module to extract the short-range and long-range correlations by stacked layers with dilated factors. STGSA (*Wei et al., 2023*) proposed a specialized graph aggregation to capture spatial-temporal dynamics.

To summarize, most of the studies adopted separate components to learn spatial and temporal correlations, failing to achieve synchronous modeling. A few works implemented simultaneous aggregation of spatial and temporal features, but their models only took into account the hidden dependencies of traffic data itself, failing to model the impacts of external factors. In addition, their spatial-temporal synchronous graphs represented only a single correlation in terms of space and time, with a failure to model spatial and temporal dynamics from multiple perspectives, resulting in the overlook of certain dependencies. To address these shortcomings in spatial-temporal synchronous modeling, this article proposes a novel approach that learns the influence of external factors by fusing them with traffic features through attention mechanisms, and constructs diverse spatial-temporal synchronous graphs to extract spatial-temporal features in multiple ways.

## PRELIMINARY

A traffic road network containing $N$ sensors can be described as a graph $\mathcal{G} = (V, E_d, A_{Ne})$, where $v_i \in V$ denotes the $i$-th node in graph $\mathcal{G}$, and each element of $E_d$ denotes an undirected edge between two nodes. The structure of $\mathcal{G}$ is represented by a spatial adjacency matrix $A_{Ne} \in \mathbb{R}^{N \times N}$, which is determined by the distribution of nodes. Each element $A_{Ne(i,j)}$ of $A_{Ne}$ denotes the connection between node $v_i$ and node $v_j$.

The traffic features (traffic flow, speed, occupancy) of node $v_i$ at time step $t$ are expressed as $x_i^t$, while the historical traffic features of graph $\mathcal{G}$ at $t$ can be described as:

$$X_H^t = \left[ x_1^t, x_2^t, \ldots, x_N^t \right]. \tag{1}$$

The task of traffic prediction is to learn a multivariant regression function $f(\cdot)$ with parameters $\theta$ for forecasting future traffic features $\hat{Y}$ based on historical traffic features $X_H$, which can be defined as:

$$\hat{Y} = f(X_H; \theta), \tag{2}$$

where $X_H = \left[ X_H^{t-T+1}, X_H^{t-T+2}, \ldots, X_H^t \right]$, $T$ denotes the length of historical traffic features. The aim of this task is to obtain the optimal parameters $\theta^*$ to minimize the error between the prediction and the ground truth, which can be formulated as:

$$\theta^* = \mathrm{argmin} L\left( Y, \hat{Y} \right), \tag{3}$$

where $Y = \left[ Y^{t+1}, Y^{t+2}, \ldots, Y^{t+T'} \right]$ is the ground truth, $T'$ denotes the length of future traffic features, and $L$ is the loss function.

## METHODOLOGY

The overall architecture of our model is presented in Fig. 2. We design FAM to take into account the influence of external factors on traffic prediction. Specifically, traffic features and external factors are first concatenated, then fused through spatial and temporal attention mechanisms, and finally mapped to a high-dimensional space and used as inputs to the model. FAM introduces external factors as part of inputs to the model and enhances the representation of traffic features, which addresses the first limitation mentioned in "Introduction". Then, we stack several dynamic multiple-graph spatial-temporal synchronous aggregation layers (DMSTSAL) to model spatial-temporal dependencies. In each layer, we construct two kinds of dynamic spatial-temporal synchronous graphs in parallel: trend spatial-temporal synchronous graph and pattern spatial-temporal synchronous graph. We then deploy a spatial-temporal synchronous aggregation module (STSAM) for each spatial-temporal synchronous graph. The outputs of the two types of modules are fused through a gating mechanism. The design of dynamic multiple graphs instead of a single graph empowers the model to learn spatial-temporal correlations comprehensively, which overcomes the second limitation mentioned in "Introduction". In addition, dilated gated convolution module (DGCM) is also designed to extract long-term dependencies in each layer, and its output is integrated with the outputs of both types of

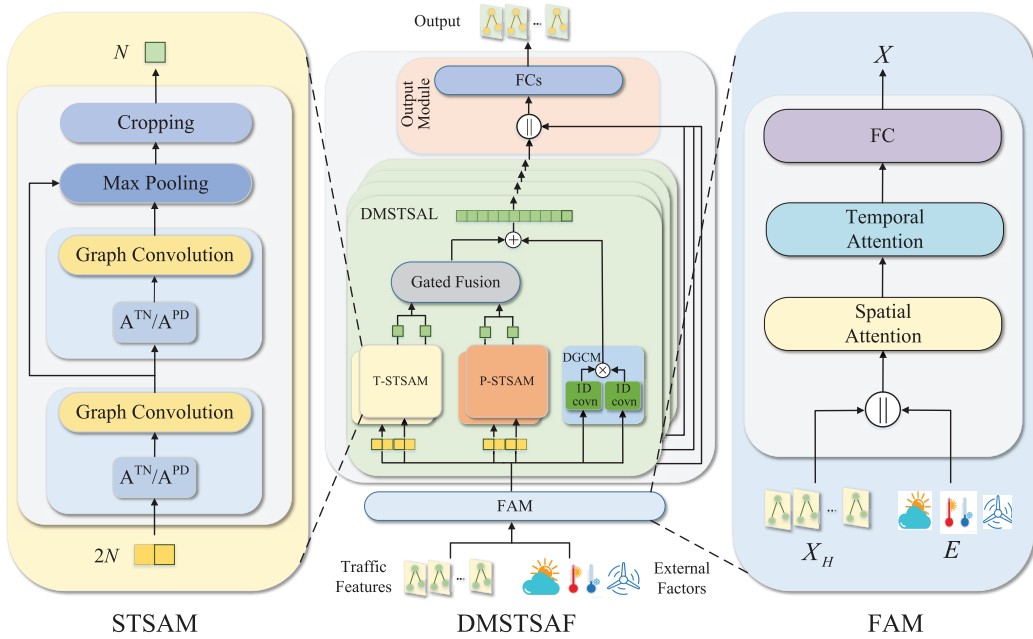

**Figure 2** **Detailed framework of DMSTSAF, FAM, and STSAM.** A DMSTSAF contains an FAW, four DMSTSALs, and an output module. Traffic features and external factors are first concatenated, then spatial attention and temporal attention are applied to enhance representation, and then high-dimensional hidden features are obtained *via* a fully connected layer. Independent T-STSAMs and parallel P-STSAMs in each DMSTSAL are designed to model spatial-temporal dependencies from multiple perspectives. In STSAMs, stacked graph convolutions followed by the max pooling and the cropping operation are incorporated with spatial-temporal synchronous graphs to extract spatial-temporal features synchronously. Gated 1D convolutions with shared parameters are utilized to learn long-term correlations in DGCM. Finally, the output module yields the predictions of the model.

STSAMs. We further design the output module with concatenation and fully connected layers to transform the outputs of DMSTSALs into predictions.

In the next subsections, we describe in detail the various components of DMSTSAF, including FAW, graph construction, STSAM, DTCM, DMSTSL, and output module. To facilitate the understanding of this study, we explain the definitions of some notations used throughout this article in Table 1.

## Feature augmentation module

Traffic features are affected by external factors such as weather, temperature, wind, *etc.* In order to integrate external factors with traffic features for accurate traffic prediction, we design FAW, whose structure is shown in Fig. 2. The comfort of the temperature is closely related to people's willingness to travel, and the traffic flow in the road network shows varying intensities under different weather conditions (*e.g.*, sunny, rainy). In addition, wind also affects people's travel plans. Therefore, we take account of the effects of four external factors that have the most impact on traffic, namely, maximum temperature, minimum temperature, weather, and wind. The external factors matrix is denoted as $E \in \mathbb{R}^{T \times N \times 4}$. The historical traffic feature matrix is denoted as $X_H \in \mathbb{R}^{T \times N \times C}$, where $C$ is

**Table 1 The definitions of some notations.**

| Notations | Explanations |
|---|---|
| $X_H$ | Historical traffic features |
| $E$ | External factors |
| $S$ | Spatial attention matrix |
| $U$ | Temporal attention matrix |
| $X$ | Fused feature |
| $G^{TN}$ | Trend spatial-temporal synchronous graph |
| $G^{PD}$ | Pattern spatial-temporal synchronous graph |
| $A^{TN}$ | Dynamic adjacency matrix of $G_{TN}$ |
| $A^{PD}$ | Dynamic adjacency matrix of $G_{PD}$ |
| $O^{TN}$ | Output of T-STSAM |
| $O^{PD}$ | Output of P-STSAM |
| $O^F$ | Output of gated fusion |
| $O^{GC}$ | Output of gated 1D convolution |
| $M^l$ | Number of step pairs in the $l$-th DMSTSAL |
| $X^l$ | Output of the $l$-th DMSTSAL |

the dimension of traffic features. To better illustrate the point in spatial and temporal attention, we define a fully connected layer as:

$$f(x) = ReLU(xW + b), \tag{4}$$

where $W$ and $b$ are learnable parameters, and $ReLU$ is the activation function.

$S \in \mathbb{R}^{T \times N \times N}$ represents the spatial attention matrix of $N$ nodes in $T$ time steps. We consider both traffic features and external factors to measure the spatial attention of different nodes. To be specific, we concatenate traffic features with external factors, and calculate the spatial correlation coefficient between node $v_i$ and $v$ at time step $t_j$ based on the scaled dot-product, which can be formulated as:

$$\alpha_{v_i,v}^{t_j} = \frac{\langle f_{s,1}(x_{v_i,t_j} \| x_{v_i,t_j}), f_{s,2}(x_{v_i,t_j} \| x_{v_i,t_j}) \rangle}{\sqrt{C+4}}, \tag{5}$$

where $\langle \cdot, \cdot \rangle$ represents the inner product, $\|$ denotes the concatenation, $f_{s,1}(\cdot)$ and $f_{s,2}(\cdot)$ represent two different fully connected layers respectively, $x_{v_i,t_j}, x_{v,t_j} \in X_H$, and $e_{v_i,t_j}, e_{v,t_j} \in E$. The spatial attention coefficient is then obtained by normalizing $\alpha_{v_i,v}^{t_j}$ via softmax:

$$s_{v_i,v}^{t_j} = \frac{\exp\left(\alpha_{v_i,v}^{t_j}\right)}{\sum_{v \in V} \exp\left(\alpha_{v_i,v}^{t_j}\right)}. \tag{6}$$

$U \in \mathbb{R}^{N \times T \times T}$ denotes the temporal attention matrix of $T$ time steps among $N$ nodes. We take into consideration both traffic features and external factors to learn the temporal

attention of various time steps. Specifically, we first concatenate traffic features with external factors, and then employ the scaled dot-product to compute the temporal correlation coefficient. After that, the temporal attention coefficient is obtained by softmax, which can be described as:

$$\begin{aligned}
\beta_{t_j,t}^{v_i} &= \frac{\langle f_{u,1}(x_{v_i,t_j} \| e_{v_i,t_j}), f_{u,2}(x_{v_i,t} \| e_{v_i,t}) \rangle}{\sqrt{C+4}}, \\
u_{t_j,t}^{v_i} &= \frac{\exp(\beta_{t_j,t}^{v_i})}{\sum_{t \in T} \exp(\beta_{t_j,t}^{v_i})},
\end{aligned}$$
(7)

where $\beta_{t_j,t}^{v_i}$ denotes the temporal correlation coefficient of node $v_i$ between time step $t_j$ and $t$, $u_{t_j,t}^{v_i}$ is the temporal attention coefficient, $f_{u,1}(\cdot), f_{u,2}(\cdot)$ represent two independent fully connected layers respectively.

After obtaining the spatial attention matrix $S$ and the temporal attention matrix $U$, traffic features and external factors are fused and mapped to the high-dimensional space, and the mathematical expression can be defined as:

$$X = f_x \left( \left( U \cdot (S \cdot (X_H \| E))^T \right)^T \right)$$
(8)

where $X \in \mathbb{R}^{T \times N \times D}$ is the fused feature.

## Dynamic spatial-temporal synchronous graph construction

Traffic features are related in multiple ways over time and space. On the one hand, nodes in different locations have adjacency and distance relationships. On the other hand, different traffic features may have the same temporal pattern or trend. A single spatial or temporal graph can only focus on one aspect of spatial-temporal dependencies while ignoring others. Aiming to model spatial-temporal correlations from multiple perspectives, we first propose two spatial graphs, *i.e.*, the neighbor graph $G^{Ne}$ and the distance graph $G^{Di}$, as well as two temporal graphs, *i.e.*, the trend graph $G^{Tr}$ and the pattern graph $G^{Pa}$, and then design two dynamic spatial-temporal synchronous graphs based on them, *i.e.*, the trend spatial-temporal synchronous graph $G^{TN}$ and the pattern spatial-temporal synchronous graph $G^{PD}$.

**Neighbor graph.** $G^{Ne}$ denotes the spatial adjacency of nodes, and the elements of its adjacency matrix $A^{Ne}$ are defined as:

$$A_{ij}^{Ne} = \begin{cases} 1, & \text{if } v_i \text{ connects to } v_j, \\ 0, & \text{else.} \end{cases}$$
(9)

**Distance graph.** $G^{Di}$ represents the distance relationship between nodes, the elements of $A^{Di}$ are fomulated as:

$$A_{ij}^{Di} = \begin{cases} \exp\left(-\frac{d_{ij}^2}{\sigma^2}\right), & \text{if } d_{ij} \le \varepsilon_D, \\ 0, & \text{else,} \end{cases}$$
(10)

where $d_{ij}$ denotes the Euclidean distance between node $v_i$ and $v_j$, $\sigma$ denotes the standard deviation of Euclidean distance, $\varepsilon_D$ is hyperparameters used to control the sparsity of $A^{Di}$.

**Trend graph.** In order to capture the trend similarity between traffic features, we propose the trend graph $G^{Tr}$ based on dynamic time warping (DTW) algorithm. Given two sequences $P = [p_1, p_2, \ldots, p_m]$ and $Q = [q_1, q_2, \ldots, q_n]$, the element $M(i, j)$ of distance matrix $M_{m \times n}$ can be calculated by $|p_i - q_j|$ (*Li & Zhu, 2021*), and the DTW distance between $P$ and $Q$ can be obtained by the iteration of the following equations:

$$
\begin{aligned}
M_D(i,j) &= M(i,j) + M_{min}(i,j), \\
M_{min}(i,j) &= min(M_D(i-1,j), M_D(i,j-1), M_D(i-1,j-1)).
\end{aligned}
\tag{11}
$$

The adjacency matrix $A^{Tr}$ can be formulated as:

$$
A^{Tr}_{ij} = \begin{cases} 1, & \text{if } M_D(i,j) \leq \varepsilon_T, \\ 0, & \text{else,} \end{cases}
\tag{12}
$$

where $M_D(i,j)$ come from (11), $\varepsilon_T$ denotes the hyperparameter to control the sparsity of $A^{Tr}$.

**Pattern graph.** Some traffic nodes exhibit strong linear correlation in their features because they are located in the same functional area (*e.g.*, residential area, commercial area, *etc.*), as shown in Fig. 1B. For the purpose of extracting pattern similarity, we design the pattern graph $G^{Pa}$ by Pearson correlation coefficient. The traffic feature series of nodes $v_i$ and $v_j$ are denoted as $X_i = \left[x_i^1, x_i^2, \ldots, x_i^T\right], X_j = \left[x_j^1, x_j^2, \ldots, x_j^T\right]$, respectively. The Pearson correlation coefficient between $v_i$ and $v_j$ can be defined as:

$$
\rho_{ij} = \frac{\sum_{t=1}^{T} \left(x_i^t - \overline{x_i}\right)\left(x_j^t - \overline{x_j}\right)}{\sqrt{\sum_{t=1}^{T} \left(x_i^t - \overline{x_i}\right)^2} \sqrt{\sum_{t=1}^{T} \left(x_j^t - \overline{x_j}\right)^2}},
\tag{13}
$$

and the adjacency matrix $A^{Pa}$ of $G^{Pa}$ can be formulated as:

$$
A^{Pa}_{ij} = \begin{cases} \rho_{ij}, & \text{if } \rho_{ij} \geq \varepsilon_P, \\ 0, & \text{else,} \end{cases}.
\tag{14}
$$

where $\varepsilon_P$ denotes the hyperparameter to manipulate the sparsity of $A^{Pa}$.

**Dynamic spatial-temporal synchronous graphs.** The spatial and temporal correlations of traffic features exist simultaneously, and to model this intricate spatial-temporal dependencies synchronously from multiple perspectives, we propose the trend spatial-temporal synchronous graphs $G^{TN}$ and the pattern spatial-temporal synchronous graphs $G^{PD}$ inspired by STSGCN (*Song et al., 2020*) and STFGNN (*Li & Zhu, 2021*). First, we design two predefined spatial-temporal synchronous graphs $G^{TN}_{Pre}$ as well as $G^{PD}_{Pre}$, each of which contains two time steps, and the adjacency matrices are denoted as $A^{TN}_{Pre} \in \mathbb{R}^{2N \times 2N}$ and $A^{PD}_{Pre} \in \mathbb{R}^{2N \times 2N}$, respectively. The structure of $A^{TN}_{Pre}$ and $A^{PD}_{Pre}$ is illustrated in Fig. 3A. The main diagonals are $[A^{Tr}, A^{Tr}]$ and $[A^{Pa}, A^{Pa}]$, separately, denoting that each node has connectivity to nodes with the same trend or pattern at the same time step, while the counter-diagonal are $[A^{Ne}, A^{Ne}]$ and $[A^{Di}, A^{Di}]$, respectively, indicating that nodes are

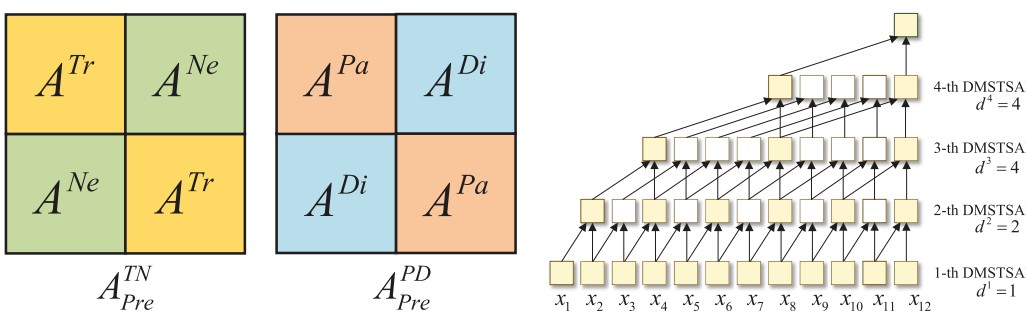

(a) The structures of the two predefined adjacency matrices     (b) The dilated step sizes in four layers

**Figure 3** The structures of the two predefined adjacency matrices and the dilated step sizes in four layers.

connected to their neighboring nodes or proximity nodes at the adjacent time step. Then, the two predefined adjacency matrices are multiplied by the learnable parameters of the same shape to obtain the adjacency matrices of the dynamic spatial-temporal synchronous graphs as follows:

$$A^{TN} = W^{TN} \otimes A^{TN}_{Pre} \in \mathbb{R}^{2N \times 2N}, A^{PD} = W^{PD} \otimes A^{PD}_{Pre} \in \mathbb{R}^{2N \times 2N}, \tag{15}$$

where $A^{TN}, A^{PD}$ denote the adjacency matrices of $G^{TN}$ and $G^{PD}$ respectively, $\otimes$ is element-wise product, $W^{TN}, W^{PD}$ are learnable parameters.

## Spatial-temporal synchronous aggregation module

Aiming to learn the hidden spatial and temporal correlations synchronously, we build spatial-temporal synchronous aggregation modules corresponding to $A^{TN}$ and $A^{PD}$, that is, the trend spatial-temporal synchronous aggregation module (T-STSAM) and the pattern spatial-temporal synchronous aggregation module (P-STSAM), which have the same architecture but with different adjacency matrices, as shown in Fig. 2. Multiple gated graph convolutions are stacked in each STSAM to learn spatial-temporal correlations simultaneously, and each gated graph convolution can be formulated as:

$$GGC(h) = tanh(hW_1^h + b_1^h) \otimes \sigma(hW_2^h + b_2^h), \tag{16}$$

where $h$ denotes the input, $tanh(\cdot)$ denotes the tanh activation function, $\sigma(\cdot)$ denotes the sigmoid activation function, and $W_1^h, W_2^h \in \mathbb{R}^{D \times D}, b_1^h, b_2^h \in \mathbb{R}^D$ are learnable parameters. The output of the previous gated graph convolution is used as the input of the next one. In addition, max pooling and a cropping operation are also included in a STSAM.

The gated graph convolution in T-STSAM allows simultaneous aggregation of information from nodes with similar temporal trends and nodes with spatial adjacencies, the mathematical equation can be formulated as:

$$h_m^{TN} = GGC(A^{TN} h_{m-1}^{TN}), \tag{17}$$

where $h_{m-1}^{TN}, h_m^{TN} \in \mathbb{R}^{2N \times D}$ denote the input and output of the $m$-th gated graph convolution in T-STSAM, respectively.

The gated graph convolution in P-STSAM implements synchronous modeling of dependencies from nodes with the same temporal pattern and nodes with short spatial distances, and the computational formula is defined as:

$$h_m^{PD} = GGC(A^{PD} h_{m-1}^{PD}), \tag{18}$$

where $h_{m-1}^{PD}, h_m^{PD} \in \mathbb{R}^{2N \times D}$ denote the input and output of the $m$-th gated graph convolution in P-STSAM, respectively.

In each STSAM, we take advantage of jump knowledge network (JK-Net) to aggregate the outputs of various gated graph convolutions and then retain the most potent representation by max pooling, which can be described as:

$$\begin{aligned}
h_{max}^{TN} &= MaxPooling(h_1^{TN}, h_2^{TN}, \ldots, h_M^{TN}), \\
h_{max}^{PD} &= MaxPooling(h_1^{PD}, h_2^{PD}, \ldots, h_M^{PD}),
\end{aligned} \tag{19}$$

where $h_{max}^{TN}, h_{max}^{PD} \in \mathbb{R}^{2N \times D}$ denote the outputs of max pooling in T-STSAM and P-STSAM, respectively, $M$ denotes the number of gated graph convolutions.

The cropping operation is performed after the max pooling to reduce the hidden features from $2N$ dimensions to $N$ dimensions, leading to the output of STSAM, which can be expressed as:

$$\begin{aligned}
O^{TN} &= h_{max}^{TN}[N:2N,:] \in \mathbb{R}^{N \times D}, \\
O^{PD} &= h_{max}^{PD}[N:2N,:] \in \mathbb{R}^{N \times D},
\end{aligned} \tag{20}$$

where $O^{TN}, O^{PD}$ denote the outputs of T-STSAM and P-STSAM, respectively.

## Dilated gated convolution module

Regarding time dimension, parallel STSAMs are conducive to extracting short-term dependencies due to their independent parameters while still lacking in modeling long-term correlations. To address this limitation, gated 1D convolution with shared parameters is utilized to extract long-term temporal features, which can be formulated as:

$$O^{GC} = tanh(\Theta_1 * X^{l-1} + b_1) \otimes \sigma(\Theta_2 * X^{l-1} + b_2), \tag{21}$$

where $O^{GC}$ denotes the output of gated 1D convolution, $X^{l-1}$ denotes the input of $l$-th layer, $\Theta_1, \Theta_2$ denote two 1D convolution respectively, and $b_1, b_2$ are learnable parameters.

Inspired by Graph WaveNet (*Wu et al., 2019*), dilated instead of fixed step sizes are introduced to $\Theta_1$ and $\Theta_2$ for expanding the receptive fields of 1D convolutions. However, the three-layer convolutions with step sizes of [1,2,4] in Graph WaveNet only cover eight historical time steps, while the length of the input sequences in our model is 12. Therefore, we deploy four layers in which the step sizes of 1D convolutions are set as [1,2,4,4] respectively, while the kernel size of 1D convolution in each layer is fixed as 2, as shown in Fig. 3B. With this set of mechanisms, the receptive field of four-layer 1D convolutions can be expanded to the length of the input time series in our model.

## Dynamic multiple-graph spatial-temporal synchronous aggregation layer

T-STSAMs and P-STSAMs with gated fusion, as well as a DGCM, compose a DMSTSAL, four DMSTSALs are stacked in a DMSTSAF, and the output of the previous layer is used as the input to the next layer, which is presented in Fig. 2. In each DMSTSAL, we first slide from the input sequence to obtain the time step pairs, and then construct two kinds of dynamic spatial-temporal synchronous graphs for every time step pair. In order to expand the receptive field and reduce the number of layers, the distances between two time steps in four layers are set to be dilated with the same step sizes as in DGCMs, which are [1,2,4,4], as shown in Fig. 3B.

Denoting the input of the $l$-th DMSTSAL as $X^{l-1} = [x_1, x_2, \ldots, x_{T^l}]$ and the distance of the two time steps as $d^l \in [1, 2, 4, 4]$, the time step pairs generated by its sliding in the $l$-th layer can be described as $[(x_1, x_{1+d^l}), (x_2, x_{2+d^l}), \ldots, (x_{T^l-d^l}, x_{T^l})]$, then the number of time step pairs can be formulated as:

$$M^l = T^l - d^l. \tag{22}$$

To achieve simultaneous learning of spatial-temporal dependencies from multiple perspectives, we construct a trend spatial-temporal synchronous graph $G^{TN}$ and a pattern spatial-temporal synchronization graph $G^{PD}$ for each time step pair. The number of each type of dynamic spatial-temporal graph is the same as the number of time step pairs, which can be obtained from (22).

Traffic features are heterogeneous across time, and to capture hidden features more accurately, we design parallel rather than shared STSMs to extract spatial-temporal dependencies. Specifically, a T-STSAM is deployed for each $G^{TN}$, and a P-STSAM is allocated for each $G^{PD}$. Thus, $M^l$ T-STSAMs and $M^l$ P-STSAMs are laid out in the $l$-th DMSTSL.

Both T-STSAMs and P-STSAMs are able to represent spatial-temporal correlations in some way, so it is necessary to fuse the outputs of two types of STSAMs. To achieve this goal, we start by aggregating the outputs of each kind of STSAM into a sequence, which can be defined as:

$$O_{AG}^{TN} = \left[O_1^{TN}, O_2^{TN}, \ldots, O_{M^l}^{TN}\right] \in \mathbb{R}^{M^l \times N \times D},$$
$$O_{AG}^{PD} = \left[O_1^{PD}, O_2^{PD}, \ldots, O_{M^l}^{PD}\right] \in \mathbb{R}^{M^l \times N \times D}, \tag{23}$$

where $O_{AG}^{TN}, O_{AG}^{PD}$ denote the outputs of T-STSAMs and P-STSAMs, respectively, they represent the results of spatial-temporal synchronous aggregation, which is the core component that forms the prediction of our model. Their elements $O_m^{TN}, O_m^{PD}, 1 \le m \le M^l$ can be obtained from (20). Then, the gating mechanism is applied to incorporate the outputs of two concatenation operations, which can be formulated as:

$$O^F = z \otimes O_{AG}^{TN} + (1 - z) \otimes O_{AG}^{PD} \in \mathbb{R}^{M^l \times N \times D}, \tag{24}$$

where $O^F$ is one of the two parts that compose the output of the current layer, $z$ is

employed to manipulate the proportion of information in the fusion, whose mathematical equation can be defined as:

$$z = \sigma\left(O_{AG}^{TN}W_1^z + O_{AG}^{PD}W_2^z + b^z\right), \tag{25}$$

where $W_1^z, W_2^z, b^z$ are learnable parameters. Another part of the current layer's outcome is the result $O^{GC}$ of the DGCM, which can be obtained from (21). The outcome $O^F$ of the fusion operation and the output $O^{GC}$ of the DGCM are added together to form the result of the $l$-th DMSTSAL, which can be formulated as:

$$X^l = O^F + O^{GC} \in \mathbb{R}^{M^l \times N \times D}, \tag{26}$$

The results of the four DMSTSALs are the inputs to the output module.

## Output module

The output module is responsible for generating the final predictions of our model, in which a concatenation that aggregates the outputs of four layers is first employed to capture comprehensive spatial-temporal correlations, which can be defined as:

$$X_{CAT} = \left(X^1 \| X^2 \| X^3 \| X^4\right) \in \mathbb{R}^{(M^1, M^2, M^3, M^4) \times N \times D}, \tag{27}$$

where $X^l, 1 \leq l \leq 4$ can be obtained from (26). A series of fully connected layers are then utilized to produce the final predictions of $T'$ time steps. Specifically, we design two-fully-connected-layers to produce the prediction of time step $t$, which can be formulated as:

$$\hat{y}_t = ReLU\left(\left(X_{CAT}W_1^t + b_1^t\right)W_2^t + b_2^t\right) \in \mathbb{R}^{1 \times N \times C}, \tag{28}$$

where $W_1^t, b_1^t, W_2^t, b_2^t$ are learnable parameters. Finally, the results of (28) repeated $T'$ times compose the final predictions of our model, which can be described as:

$$\hat{Y} = \left[\hat{y}_1, \hat{y}_2, \ldots, \hat{y}_{T'}\right] \in \mathbb{R}^{T' \times N \times C}. \tag{29}$$

Smooth L1 loss rather than L1 loss is chosen as the loss function, which deals with the unsmooth disadvantage at the zero-point and can be defined as:

$$L(Y, \hat{Y}) = \begin{cases} \frac{1}{2}(Y - \hat{Y})^2/\delta, & if \left|Y - \hat{Y}\right| \leq \delta, \\ \left|Y - \hat{Y}\right| - \frac{1}{2}\delta, & otherwise \end{cases}, \tag{30}$$

where $Y, \hat{Y}$ denote the true value and the predicted value, respectively, and $\delta$ denotes a threshold parameter to determine the sensitivity.

## EXPERIMENTS

### Datasets

Four public datasets are chosen for evaluating the prediction performance of DMSTSAF: PEMS03, PEMS04, PEMS07, and PEMS08. All datasets are generated by Caltrans Performance Measurement System (PeMS). Concretely, traffic features in these datasets

**Table 2 Detailed information of datasets.**

| Datasets | Samples | Number of nodes | Traffic features | Time span |
|---|---|---|---|---|
| PEMS03 | 26,208 | 358 | Flow | 09/01/2018–11/30/2018 |
| PEMS04 | 16,992 | 307 | Flow, Speed, Occupancy | 01/01/2018–02/28/2018 |
| PEMS07 | 28,224 | 883 | Flow, Speed, Occupancy | 05/01/2017–08/31/2017 |
| PEMS08 | 17,856 | 170 | Flow | 07/01/2016–08/31/2016 |

are collected by sensors located on California highways at 5-min intervals, which means there are 12 time steps in an hour. Spatial graphs are constructed from the distribution of sensors. The distinctions among the datasets are the geographic locations of sensors and the temporal ranges of data. The detailed information of four datasets can be found in Table 2.

## Experiment settings and evaluation metrics

In our experimental implementation, each dataset is partitioned for training, validation, and testing in a ratio of 60%, 20%, and 20%. We employ the traffic flow of 12 historical time steps (1 h) to predict the traffic flow of 12 future time steps. The traffic flow in each dataset is standardized using Z-score normalization.

DMSTSAF is performed by Pytorch using a PC with NVIDIA RTX 3080. We chose hyperparameters of our model experimentally to ensure superior performance. The sparsity of the adjacency matrices for the distance graph, the trend graph, and the pattern graph are set as 0.01. Each STSAM consists of two graph convolutions, and the output dimension of the fully connected layer in FAW and the hidden dimensions of graph convolutions in STSAM are set as $D = 64$. The hidden representations of two fully-connected layers to generate predictions are tuned as [128,1], respectively. The optimizer in experiments is set as Adam. The batch sizes for four datasets are tuned as [32,32,8,64], respectively. The initial learning rate is set as 0.003 and scaled down to 0.3 times every 30 epochs to shorten the training time. Enroll up to 200 epochs in each training.

We take three metrics to evaluate the performance of models, which are mean absolute error (MAE), mean absolute percentage error (MAPE), and root mean square error (RMSE). Smaller values indicate better performance for these metrics, and their mathematical equation can be formulated as:

$$
\begin{aligned}
MAE(y_i, \widehat{y}_i) &= \frac{1}{N \times T'} \sum_{i=1}^{N \times T'} |y_i - \widehat{y}_i|, \\
MAPE(y_i, \widehat{y}_i) &= \frac{1}{N \times T'} \sum_{i=1}^{N \times T'} \frac{|y_i - \widehat{y}_i|}{y_i}, \\
RMSE(y_i, \widehat{y}_i) &= \sqrt{\frac{1}{N \times T'} \sum_{i=1}^{N \times T'} (y_i - \widehat{y}_i)^2},
\end{aligned}
\tag{31}
$$

where $y_i$, $\widehat{y}_i$ denote the true value and the predicted value, respectively.

## Baseline methods

We compare DMSTSAF with the following seven models for traffic flow prediction:

- FC-LSTM (*Sutskever, Vinyals & Le, 2014*): Long short-term memory network with fully connected layers, a variant of the recurrent neural network, which consists of the forget gate, the input gate, and the output gate.
- GRU (*Fu, Zhang & Li, 2016*): Gate recurrent unit, a variant of LSTM, which simplifies the three gates in LSTM to two, *i.e.*, the reset gate and the update gate.
- T-GCN (*Zhao et al., 2020*): Temporal graph convolutional network, which incorporates GCN with GRU to extract spatial-temporal features.
- DCRNN (*Li et al., 2018*): Diffusion convolution recurrent neural network, in which random walks of a graph are utilized to model spatial dependence, GRU is employed to learn temporal correlation.
- STGCN (*Yu, Yin & Zhu, 2018*): Spatio-temporal graph convolution network, which takes advantage of GCN to extract spatial features, and makes use of 1D convolution rather than recurrent neural network to model temporal dependency.
- STSGCN (*Song et al., 2020*): Spatial-temporal graph convolutional network, which constructs localized spatial-temporal graphs based on spatial and temporal adjacencies, and designs spatial-temporal synchronous graph convolutional modules to learn spatial-temporal correlations synchronously.
- STFGNN (*Li & Zhu, 2021*): Spatial-temporal fusion graph neural network, an improved study based on STSGCN, which designs temporal graphs with dynamic time warping algorithm and proposes spatial-temporal fusion graphs to model localized spatial-temporal dependencies, and then designs gated CNN to extract long-range dependencies.

## Experimental results

The experiments of traffic flow prediction are performed on four datasets, and a comparison of the results for all models is shown in Table 3. DMSTSAF obtains the smallest metrics on all four datasets, demonstrating the consistent superiority of DMSTSAF over the baselines. Compared to the state-of-the-art baseline STFGNN, DMSTSAF improves 4.27%, 3.72%, 8.54%, and 7.89% in terms of MAE on PEMS03, PEMS04, PEMS07, and PEMS08 respectively, while the improvements of MAPE are 6.63%, 7.49%, 7.95%, and 7.85%. In addition, our model also achieves 3.68%, 4.23%, 8.38%, and 7.65% improvements in terms of RMSE.

Moreover, for the purpose of evaluating the ability of models on multi-step prediction, we conduct prediction experiments for 12 future time steps on PEMS03. In comparison to STFGNN, our model shows 1.11–8.89% improvement in terms of MAE, 1.81–13.19% improvement for MAPE, and 1.38–7.48% improvement for RMSE. To achieve a more intuitive comparison, we illustrate the results of each model with a line plot, as shown in Fig. 4, which verifies that DMSTSAF overwhelmingly outperforms all the baselines.

**Table 3 Results of DMSTSAF and baselines on four datasets.**

| Datasets | Metric | FC-LSTM | GRU | T-GCN | DCRNN | STGCN | STSGCN | STFGNN | DMSTSAF |
|----------|--------|---------|-----|-------|-------|-------|--------|--------|---------|
| PEMS03 | MAE | 27.35 | 27.19 | 20.83 | 21.07 | 20.29 | 18.30 | 17.33 | 16.59 |
| | MAPE (%) | 25.16 | 24.92 | 21.58 | 20.43 | 18.98 | 17.58 | 16.90 | 15.78 |
| | RMSE | 42.32 | 42.24 | 31.18 | 33.23 | 33.08 | 30.20 | 29.04 | 27.97 |
| PEMS04 | MAE | 34.33 | 34.13 | 26.02 | 27.57 | 25.37 | 22.38 | 20.14 | 19.39 |
| | MAPE (%) | 21.72 | 21.53 | 17.08 | 18.29 | 15.50 | 15.03 | 13.76 | 12.73 |
| | RMSE | 50.01 | 49.98 | 38.19 | 42.07 | 39.11 | 35.33 | 32.62 | 31.24 |
| PEMS07 | MAE | 38.33 | 37.79 | 30.37 | 31.29 | 30.91 | 25.15 | 23.76 | 21.73 |
| | MAPE (%) | 16.71 | 16.83 | 13.83 | 15.09 | 14.49 | 10.74 | 9.96 | 9.17 |
| | RMSE | 57.56 | 56.72 | 43.39 | 47.21 | 47.12 | 40.79 | 38.48 | 35.25 |
| PEMS08 | MAE | 28.90 | 28.12 | 21.37 | 21.21 | 21.39 | 17.72 | 16.99 | 15.65 |
| | MAPE (%) | 17.77 | 16.92 | 13.66 | 13.08 | 13.13 | 11.61 | 10.96 | 10.10 |
| | RMSE | 41.83 | 41.85 | 30.69 | 31.43 | 31.42 | 27.21 | 26.80 | 24.75 |

To further demonstrate the prediction performances of DMSTSAF and the best baselines, we visualize the ground-truth and predictions of STSGCN, STFGNN, and DMSTSAF on PEMS03 test set for both sunny and rainy days, as presented in Fig. 5. It can be seen that compared to the best baselines, our proposed DMSTSAF can fit the ground-truth more accurately in different weather conditions.

## Ablation study

To evaluate the validity of different modules in DMSTSAF, ablation studies are implemented on PEMS04 and PEMS08. We design six variants, a short illustration is introduced as follows:

- -FAW, which uses only traffic features as the input to the model, and utilizes an FC rather than a feature augmentation module to transform the input from $C$ dimensions to $D$ dimensions.
- $-G^{TN}$, which removes trend spatial-temporal synchronous graphs $G^{TN}$ and the corresponding T-STSAM, and retains only pattern spatial-temporal synchronous graphs $G^{PD}$ as well as P-STSAM.
- $-G^{PD}$, which is the opposite of $-G^{TN}$, removing $G^{PD}$ and P-STSAM, while retaining $G^{TN}$ as well as T-STSAM.
- -Dilation, the step sizes of 1D convolutions in DGCM and the distance between two time steps to construct dynamic spatial-temporal synchronous graphs are fixed to 1.
- -Concatenation, which removes the concatenation on the outputs of four DMSTSALs, and treats the output of the 4-th DMSTSAL as the input to the output module.
- -DGCM, which removes dilated gated convolution modules from four DMSTSALs.

Table 4 demonstrates the results of ablation studies, which illustrates that DMSTSAF outperforms variants on PEMS04 as well as PEMS08. Compared to the three indicators of

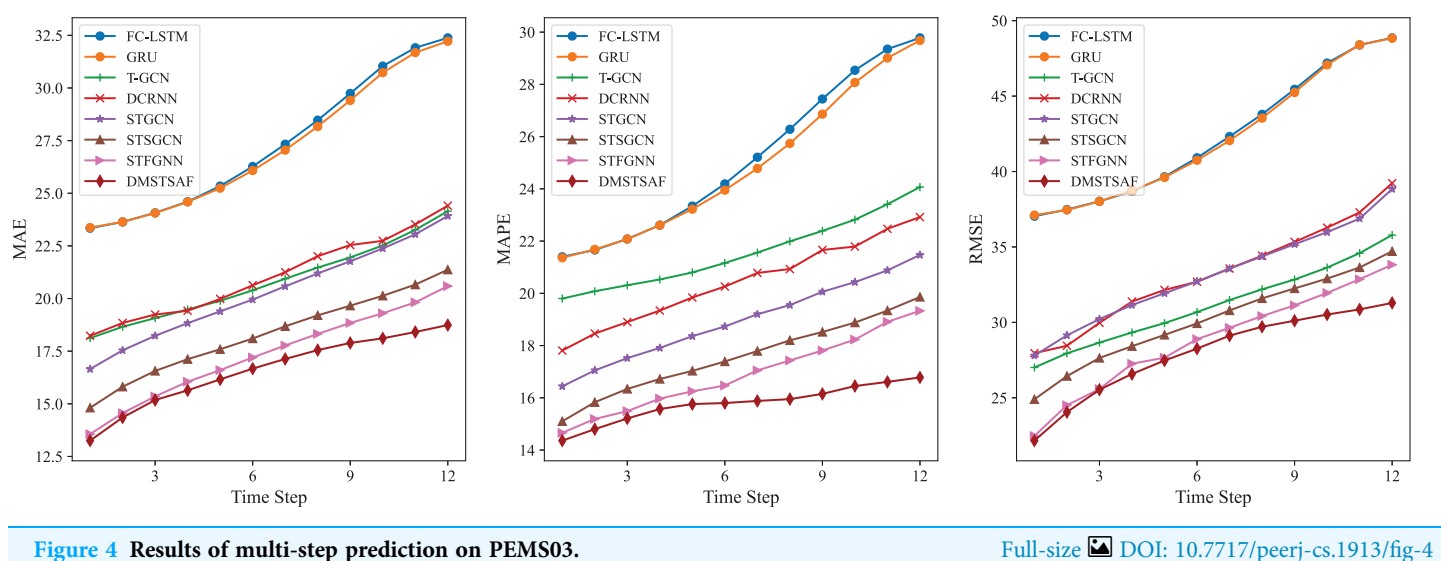

**Figure 4  Results of multi-step prediction on PEMS03.**

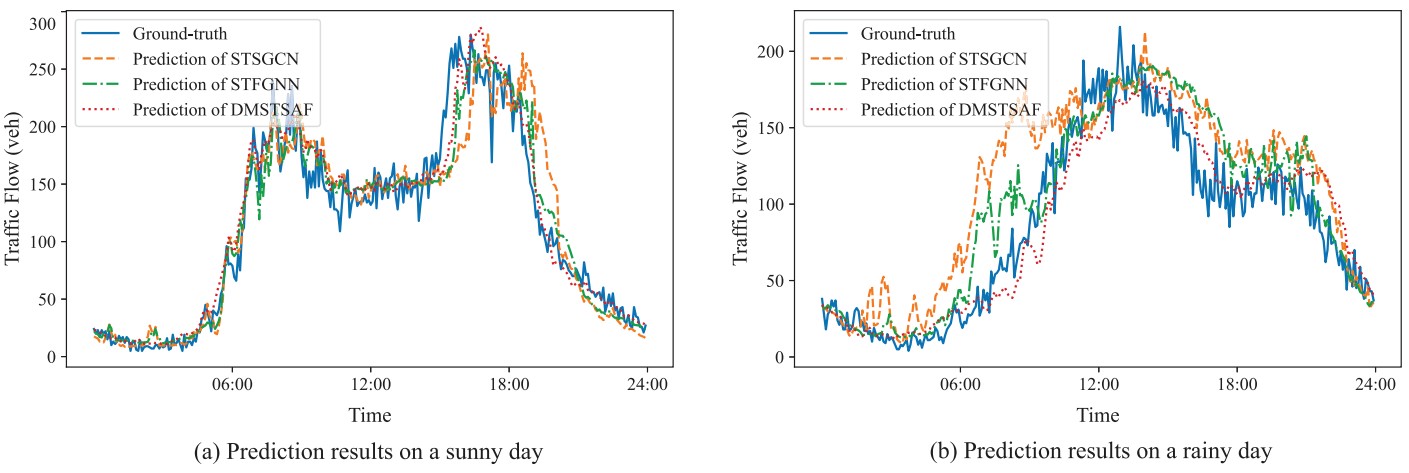

(a) Prediction results on a sunny day    (b) Prediction results on a rainy day

**Figure 5  Prediction results of STSGCN, STFGNN, DMSTSAF.**

-FAW, our model achieves 1.92%, 2.60% as well as 1.67% improvement on PEMS04. Furthermore, it improves by 3.04%, 2.79%, and 1.39% on PEMS08. These improvements indicate the effectiveness of fusing traffic features and external factors in learning spatial-temporal correlations. In contrast with $-G^{TN}$, DMSTSAF achieves 1.02%, 1.53%, 1.01% improvements on PEMS04, as well as 1.39%, 2.23%, 0.88% improvements on PEMS08. In comparison to $-G^{PD}$, improvements of 0.6%, 0.94%, 0.54% on PEMS04 are obtained by our model, while improvements of 0.70%, 1.56%, 0.52% are achieved on PEMS08. These achievements verify the validity of dynamic multiple-graph in modeling spatial-temporal dependencies. Compared to -DGCM, our model achieves 1.87%, 2.68%, as well as 2.74% improvement on PEMS04. Furthermore, it improves 2.49%, 4.54%, and 1.67% on

**Table 4 Results of ablation studies.**

| Model & Variants | PEMS04 | | | PEMS08 | | |
|---|---|---|---|---|---|---|
| | MAE | MAPE (%) | RMSE | MAE | MAPE (%) | RMSE |
| DMSTSAF | 19.39 | 12.73 | 31.24 | 15.65 | 10.10 | 24.75 |
| -FAW | 19.77 | 13.07 | 31.77 | 16.14 | 10.39 | 25.10 |
| $-A^{TN}$ | 19.59 | 12.93 | 31.56 | 15.87 | 10.33 | 24.97 |
| $-A^{PD}$ | 19.46 | 12.85 | 31.41 | 15.76 | 10.26 | 24.88 |
| -Dilation | 19.62 | 13.05 | 31.61 | 15.95 | 10.37 | 25.05 |
| -Concatenation | 20.12 | 13.35 | 32.52 | 17.01 | 10.89 | 26.67 |
| -DGCM | 19.76 | 13.08 | 32.12 | 16.05 | 10.58 | 25.17 |

PEMS08. These improvements illustrate the positive role of DGCM in capturing long-term correlations. In addition, there are also different levels of improvement in our model compared to other variants, which demonstrates the efficiencies of the corresponding components.

## Parameters study

In order to further validate our model, parameters study are implemented on PEMS04 and PEMS08. The number of graph convolutions and the hidden dimensions $D$ in STSAM are under consideration, Fig. 6 presents the results of experiments. Our model achieves minimal errors as each STSAM consists of two graph convolutions. As for the hidden dimensions, it can be found in Fig. 6 that our model obtains optimal metrics when $D$ equals to 64.

## DISCUSSION

By means of the results in "Experimental Results", it can be inferred that our proposed model outperforms the seven baselines. FC-LSTM and GRU are not satisfactory since they only take into account temporal dependency and ignore spatial relationships. T-GCN, DCRNN, and STGCN utilize graph convolution network to model spatial dependence and further employ temporal components to extract temporal features, leading to improved performance than models only applicable for time sequences. However, their separate modules for spatial and temporal modeling limit the efficiency of feature extraction. STSGCN designs synchronous graphs to characterize spatial-temporal correlations simultaneously. Further, STFGNN proposes temporal graphs generated by a dynamic time warping algorithm, and then designs spatial-temporal fusion graphs. Although they both achieve synchronous learning compared to previous work, their performances are still insufficient. First, neither of them takes account of the impact of external factors, which weakens the ability of models to extract comprehensive dependencies. Second, their single spatial-temporal graph structures result in overlooking certain potential correlations. Our model integrates maximum temperature, minimum temperature, weather, as well as wind with traffic features by spatial and temporal attention, and then feeds the fused features into subsequent modules. Therefore the influence of external factors is taken into

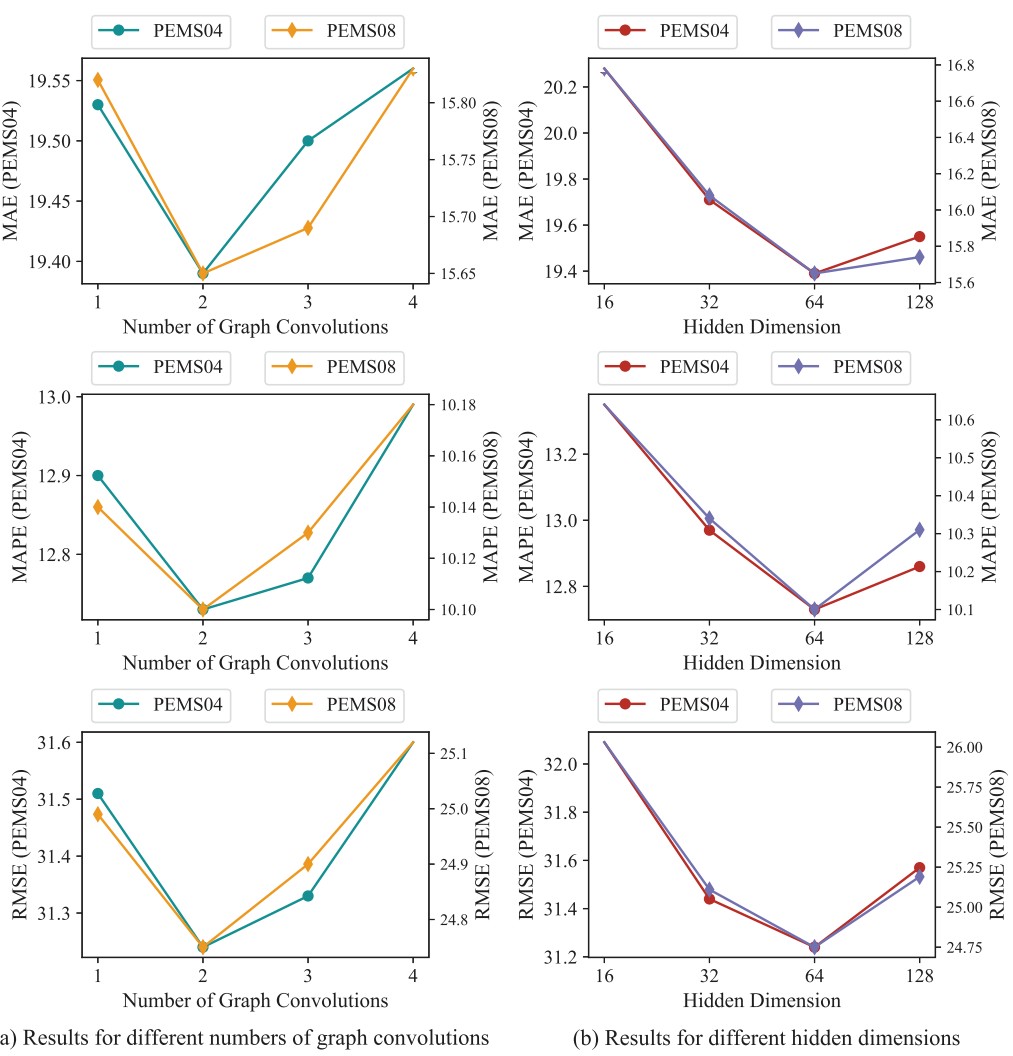

(a) Results for different numbers of graph convolutions    (b) Results for different hidden dimensions

**Figure 6  Experimental results of parameters on PEMS04 and PEMS08.**

consideration in spatial-temporal modeling. In addition, we introduce four graphs that lead to two dynamic spatial-temporal synchronous graphs, bringing about multi-perspective spatial-temporal simultaneous modeling of hidden features, enhancing the proposed model's performance for learning spatial and temporal dependencies.

Ablation studies in "Ablation Study" illustrate the effectiveness of various components in our model. FAW is effective for feature extraction because traffic features vary significantly with different external factors. Both $G^{TN}$ and $G^{PD}$ contribute noticeably to the improved performance of DMSTSAF due to their capacity for learning spatial-temporal correlations from different perspectives. Other components are also beneficial to strengthening the model's performance. Specifically, dilated step sizes expand the receptive fields of stacked layers, the concatenate operation on outputs of four layers retains more useful hidden information, and DGCM captures long-term temporal dependence.

Furthermore, the excellent performance of DMSTSAF benefits from the appropriate number of graph convolutions and the suitable hidden dimensions $D$ in STSAM, which is indicated in "Parameters Study". When each STSAM comprises only one graph convolution, this leads to the inefficient representation of hidden dimensions, while three or four graph convolutions consisting of one STSAM result in over-smoothing. If we tuned hidden dimensions in STSAM to 16 or 32, the inability to obtain efficient deep representations restricts the capacity to model dependencies, but tuning hidden dimensions to 128 results in over-fitting.

Although diverse experiments confirm the effectiveness of our model, there are still some shortcomings. First, the proposed method cannot efficiently deal with the missing data in datasets. Second, the external factors only include meteorological conditions, and further extensions are yet to be made. In future work, we plan to introduce processing algorithms for missing data and incorporate additional external factors to improve our model further.

## CONCLUSION

In this article, a novel dynamic multiple-graph spatial-temporal synchronous aggregation framework is proposed for traffic prediction, which incorporates traffic features with external factors *via* spatial-temporal attention as the input, thus modeling the impact of external factors on traffic features. Meanwhile, it characterizes spatial-temporal dependencies from multiple perspectives through two kinds of dynamic spatial-temporal synchronous graphs. In addition, two types of spatial-temporal synchronous aggregation modules empower the model to extract spatial-temporal features synchronously, and the dilated step sizes expand the receptive field. Finally, the many-to-one mechanism in the output module transforms hidden features into accurate predictions. Extensive experiments are implemented on four real-world datasets, and numerical results demonstrate that the proposed model consistently outperforms all baselines, achieving average improvements ranging from 5.99–7.48% on three metrics compared to the state-of-the-art study. In the future, we will attempt to deal with missing data in datasets and take into account additional external factors, such as traffic accidents, to improve our model's performance further.

### Funding

This work was supported by the National Natural Science Foundation of China (No. 61771265), the Funding for the 6th "333 Talent" Project in Jiangsu Province (No. 2022044), the Postgraduate Research & Practice Innovation Program of Jiangsu Province (No. KYCX22_3341, KYCX23_3396). The funders had no role in study design, data collection and analysis, decision to publish, or preparation of the manuscript.

## Grant Disclosures

The following grant information was disclosed by the authors:

National Natural Science Foundation of China: 61771265.

6th "333 Talent" Project in Jiangsu Province: 2022044.

Postgraduate Research & Practice Innovation Program of Jiangsu Province: KYCX22_3341, KYCX23_3396.

## Competing Interests

The authors declare that they have no competing interests.

## Author Contributions

- Xian Yu conceived and designed the experiments, performed the experiments, performed the computation work, prepared figures and/or tables, authored or reviewed drafts of the article, and approved the final draft.
- Yinxin Bao performed the experiments, analyzed the data, authored or reviewed drafts of the article, and approved the final draft.
- Quan Shi conceived and designed the experiments, analyzed the data, authored or reviewed drafts of the article, and approved the final draft.

## Data Availability

The source code and raw data of the network DMSTSAF are available in the Supplemental File and at GitHub: https://github.com/yxcos/DMSTSAF.

## Supplemental Information

Supplemental information for this article can be found online at http://dx.doi.org/10.7717/peerj-cs.1913#supplemental-information.

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
