# Peer review of "Dynamic multiple-graph spatial-temporal synchronous aggregation framework for traffic prediction in intelligent transportation systems"

_PeerJ Computer Science, doi:10.7717/peerj-cs.1913_

## Round 0.1 · original submission · Major Revisions

The reviewers have some concerns about the work. Please improve the work according to the comments. Then it will be evaluated again.

**Language Note:** PeerJ staff have identified that the English language needs to be improved. When you prepare your next revision, please either (i) have a colleague who is proficient in English and familiar with the subject matter review your manuscript, or (ii) contact a professional editing service to review your manuscript. PeerJ can provide language editing services - you can contact us at copyediting@peerj.com for pricing (be sure to provide your manuscript number and title). – PeerJ Staff

Reviewer 1 ·

Basic reporting

This manuscript proposes a novel dynamic multi-graph spatial-temporal synchronous aggregation framework for traffic prediction, which describes spatial-temporal dependencies from multiple perspectives through two kinds of dynamic spatial-temporal synchronization graphs and constructs an aggregation module to extract the spatial-temporal features, and which also considers the effects of external factors on traffic features. Eventually the framework achieves optimal results on all four public datasets. This work provides a reference for constructing spatial-temporal synchronization maps for traffic prediction from multiple perspectives, but there are still the following problems to be solved.
In the Feature augmentation module of the Methodology, please further explain why the four characteristics of maximum temperature, minimum temperature, weather and wind were used?
In the Dynamic spatial-temporal synchronous graph construction of the Methodology, why is the pattern graph in the temporal graph calculating the Pearson coefficient between two nodes?
In Fig.3a, A_pre^TN and A_pre^PD should be labeled in the figure.
In the Dilated gated convolution module of the Methodology, author should give the basis for the hyperparameter settings of DGCM.
In the Ablation study of the Experiments, the authors should add experiments that remove the DGCM rather than modifying its parameters to verify the validity of the DGCM.

Experimental design

-

Validity of the findings

-

Additional comments

-

Reviewer 2 ·

Basic reporting

Author presented the manuscript "Dynamic multiple-graph spatial-temporal synchronous aggregation framework for traffic prediction". Here some of the points need to be clear from the authors.
In title Traffic prediction was mentioned using some aggregation framework. What purpose traffic prediction was proposed by author ? suppose its for transportation system means, author can include this keywords in the title as well as author can give some introduction in the abstract as well as survey sections.

Author used many short forms without proper abbreviations like AST-GCN, STSGCN, STFGNN, DSTSGN and GMAN. Author should abbreviate these terms first usage.

Experimental design

In Figure.2, Author can interrelate the figures (a) DMSTSAF architecture, (b) FAW architecture and (c) STSAM architecture in terms of external factors. All the figures are inter-related means , author can combined all the figures as single one and describe the flow model for this proposed DMSTSAF. How the traffic features can be extracted ?. Similarly what is methods applied for id conversion in DGCM ?

Validity of the findings

How Distance Graph can be defined ? what is the parameter for spatial distance ? Why author applied two spatial graphs as neighbor graph and spatial graph ?
how the equation 23 to 29 related to experimental settings ? How these equations were helped to find out the outcome of traffic predictions? I think author might missed important mathematical formula related to aggregation framework. So better author can refer recent manuscript related to this and find the solutions. Finally proofread should be done during revision.

---

## Round 0.2 · accepted · Accept

Thanks to the authors for your efforts to improve the work. To the best of my knowledge, I believe you successfully revised the paper accordingly. Thus, I recommend the acceptance of the current version.